# Socio-cultural and structural barriers influencing parents' knowledge and access to information on schistosomiasis in children around Ugandan Lakes

Lisa Sophie Reigl [1]*, Maxson Kenneth Anyolitho[2], Stella Neema[3], Mary Amuyunzu-Nyamongo[4], Andrea Buhl[1,5], Jennifer Burrill[6], Marie Frese[1,7], Djouquou Alexise Gnahore[8], Eveline Hürlimann[9,10], Lobohon Suzanne Lavry Épouse Yao[8], Janet Masaku[11], Nora Monnier[9,10], Ashley Preston[6], Alice Sereti Sinkeet[4], Peter Steinmann[9,10], Alain Toh[8], Orkan Okan[12,13], Andrea S. Winkler[1,14,15], Isabelle L. Lange[1,16]

1 Department of Neurology, TUM University Hospital and Center for Global Health, School of Medicine and Health, Technical University of Munich, Munich, Germany, 2 Department of Community Health, Faculty of Public Health, Lira University, Lira, Uganda, 3 Department of Sociology and Anthropology, Makerere University, Kampala, Uganda, 4 Department of Research and Programmes, African Institute for Health and Development (AIHD), Nairobi, Kenya, 5 Institute of Social Anthropology, University of Basel, Basel, Switzerland, 6 Unlimit Health, London, United Kingdom, 7 Institute for Occupational and Maritime Medicine, University Medical Center Hamburg-Eppendorf, Hamburg, Germany, 8 Département de Sociologie, Université Félix Houphouet-Boigny, Abidjan, Côte d'Ivoire, 9 Swiss Tropical and Public Health Institute, Allschwil, Switzerland, 10 University of Basel, Basel, Switzerland, 11 Eastern and Southern Africa Centre of International Parasite Control (ESACIPAC), Kenya Medical Research Institute (KEMRI), Nairobi, Kenya, 12 Professorship of Health Literacy, School of Medicine and Health, Technical University of Munich, Munich, Germany, 13 WHO Collaborating Centre for Health Literacy, Department of Health and Sport Sciences, School of Medicine and Health, Technical University of Munich, Munich, Germany, 14 Department of Community Medicine and Global Health, Institute of Health and Society, University of Oslo, Oslo, Norway, 15 Department of Global Health and Social Medicine, Harvard Medical School, Massachusetts, United States of America, 16 Department of Infectious Disease Epidemiology and International Health, London School of Hygiene & Tropical Medicine, London, United Kingdom

☙ These authors contributed equally to this work.

* lisa.reigl@tum.de

## Abstract

### Background

In Uganda, the national prevalence of schistosomiasis is 36.1% among the two-to-four-year age group. Knowledge about schistosomiasis and socio-cultural factors have been reported to influence adult participation in mass drug administrations of praziquantel, the standard medication used to treat the disease. In advance and support of the (pilot) introduction of the recently developed paediatric praziquantel formulation (arpraziquantel), we conducted research on parent and other community members' knowledge about paediatric schistosomiasis, their information sources, and potential perceived barriers that influence knowledge and access to information on schistosomiasis.

**Data availability statement:** Data from interviews cannot be made publicly available due to confidentiality, privacy, and ethical concerns. The qualitative data includes sensitive information that could identify participants or compromise their privacy. Moreover, the data was collected under an agreement of confidentiality with the participants, ensuring that their identities and the details shared would remain protected. We are committed to the principles of transparency and openness in research within the constraints of these ethical and legal obligations. Anonymized excerpts relevant to the study can be requested via neurologie@mri.tum.de.

**Funding:** This work was supported by the GHIT Fund (G2020-102 to LSR, MKA, SN, AB, JB, MF, EH, NM, AP, PS, ASW, ILL) and EDCTP (RIA2019-2895 to MAN, DAG, SL, ASS, AT). The authors JM and OO received no specific funding for this work. The Pediatric Praziquantel Consortium is financially supported by Merck, with in-kind contributions from partners and grants from the Bill & Melinda Gates Foundation (2012)(OPP1063223), the Global Health Innovative Technology Fund (GHIT) (G2013-212, G2014-206, G2016-110, G2018-210 & G2020-102) (https://www.ghitfund.org/ ), and the European & Developing Countries Clinical Trials Partnership (EDCTP) (RIA2016S-1641& RIA2019-2895) (https://www.edctp.org/). The funders had no role in the study design, data collection and analysis, preparation of the manuscript or the decision to publish.

**Competing interests:** The authors have declared that no competing interests exist.

## Methods

This cross-sectional qualitative study consisted of 65 in-depth interviews and ten focus group discussions with parents/guardians of preschool-aged children (PSAC), Village Health Teams (VHTs), health professionals, community leading persons and district officials, in addition to direct observations. We collected data concurrently in Hoima District at Lake Albert and Bugiri at Lake Victoria, Uganda, in 2022. The analysis followed both an inductive and a deductive thematic approach.

## Results

Despite high parental awareness of the disease and knowledge of signs and symptoms, we identified less familiarity with its transmission and prevention. With limited communication channels, VHTs emerged as the primary information sources but reflected varied local understandings of schistosomiasis. Parents expressed a desire to be informed about the new paediatric treatment through VHTs, health professionals, and community leaders, and partly through the radio. Other mentioned socio-cultural and structural barriers to information flow included language barriers, illiteracy, high population movement, fears, and resentment from restrictive government fishing laws.

## Conclusions

Carefully designed information campaigns tailored to local circumstances and health literacy needs should be carried out by trained VHTs and preferably supported by health professionals and the local leadership structure. These components are essential to inform parents/guardians of PSAC, enabling them to make well-informed decisions for their children's health.

## Author summary

The studied lakeside communities in Uganda have limited knowledge about the transmission and prevention of schistosomiasis while simultaneously being aware of the significance of the disease. This study was conducted in preparation for the introduction of a paediatric praziquantel formulation and an associated community sensitisation campaign. Therefore, we explored parents' knowledge of the disease in preschool-aged children and their preferred information channels through interviews, focus group discussions and observations. We talked to various other stakeholders important to the community or involved in health campaigns. Due to the remoteness of the studied regions, broadcast media such as television were barely named as information sources. Instead, parents relied on community health workers, health professionals and the village leaders. Newspapers or other written material did not play a role, given that the

population was highly illiterate. Moreover, high population movements (fisher communities, refugees) and the use of several languages pose challenges to continuous health information. We identified several fears, for example that mass treatments could be a hidden population reduction strategy. Considering these findings, we suggest that control programmes and their information campaigns take local socio-cultural and structural factors carefully into account and be adjusted to support health programmes effectively.

## 1. Introduction

Schistosomiasis is a disease caused by parasitic flatworms that affected around 237 million people worldwide in 2019 [1]. In terms of the socioeconomic impact of the disease, schistosomiasis is considered the second most important parasitic infection [2], causing mortality and morbidity, loss of productivity and financial burdens related to medical costs. Endemic areas have been documented in South America, the Caribbean, Asia and Africa [1]. Schistosomiasis exists in almost all sub-Saharan African (SSA) countries, and a higher likelihood of infection is seen among populations who work with natural water bodies, and among women and children [3,4]. The disease can cause rash, fever, respiratory symptoms, diarrhoea, abdominal pain, and bloody urine. It can have severe long-term consequences such as malnutrition, chronic anaemia, infertility, and liver and kidney damage [5,6]. In children, if not treated, the disease can lead to malnutrition and growth stunting [5,7], chronic anaemia [5,7,8], impaired development in childhood due to prolonged systematic and organ-specific inflammation [5,7], impaired neurological and cognitive functioning in children [8], as well as educational, learning and memory deficits [9,10]. In adults, signs/symptoms and long-term effects are comparable, yet in children, the impacts are more severe due to their disruption of healthy child development during a critical stage.

In Uganda, the national prevalence of schistosomiasis is 25.6%, with the highest prevalence of 36.1% among the two-to-four-year age group [11]. In five endemic districts along Lake Victoria, the prevalence among children aged one to five was 39.3% [12]. People or animals infected with schistosomes transmit schistosomiasis through contamination of fresh-water with faeces or urine containing parasite eggs. Those eggs hatch in the water and release immature larvae, which penetrate snails where they undergo further larval development. People become infected when larvae, that are released from freshwater snails, penetrate the skin during contact with the contaminated water. In the body, these larvae mature into adult worms and live in blood vessels around the bladder or the intestine, where females lay eggs after mating. Some eggs exit the body through faeces or urine, continuing the life cycle of schistosomiasis, while others become lodged in tis-sues, triggering immune responses that can lead to chronic inflammations and subsequent organ damage [1]. To eliminate schistosomiasis, the World Health Organization's (WHO) strategy involves preventive chemotherapy with praziquantel for at-risk populations, access to clean drinking water, improved sanitation, hygiene education, environmental management, and snail control [1].

Despite extensive efforts, Uganda did not reach its previous goal of eliminating schistosomiasis as a public health problem by 2020 [13]. These efforts included mass drug administrations (MDAs) with praziquantel, focusing on high-prevalence areas, and ongoing distribution since 2003 to children aged five and above and sometimes also adults [14]. Uganda – like other countries – has consistently fallen short of achieving the WHO's target of covering 75% of school-aged children and adults at risk with praziquantel through MDA [15–17], which is the supporting pillar in the country's mechanisms to control schistosomiasis. Additionally, the current strategy in Uganda, mirroring practices in other nations, excludes mass treatment of preschool-aged children (PSAC) due to the unsuitability of the current praziquantel because of its large tablet size, bitter taste and dosage [18,19].

To address the treatment gap, a paediatric-child-friendly formulation of praziquantel (arpraziquantel, 150mg) was devel-oped by the Pediatric Praziquantel Consortium, which received a positive opinion from the European Medicine Agency in December 2023 [20] and was added to the WHO list of prequalified medical products in May 2024 [21]. It has positive

attributes such as being smaller in tablet size and dose, dissolvable in water and improved in taste and was found in a clinical trial to be safe and effective [22,23]. The implementation research programme ADOPT, conducted by the Pediatric Praziquantel Consortium, addresses the appropriate distribution of arpraziquantel, focusing on aspects such as social mobilisation, community acceptance, assessment of existing schistosomiasis drug delivery strategies, and identifying and piloting suitable treatment platforms. The Ugandan government is preparing to pilot the drug distribution in two districts with the aim of evaluating the best strategies for equitable and sustainable access to this novel paediatric treatment. However, a low MDA coverage significantly reduces this method's impact in decreasing transmission and the associated health effects of schistosomiasis for the community [24].

Socio-cultural factors, including knowledge, attitudes, practices, health literacy, and personal characteristics such as age, education, sex, and economic and environmental conditions, have been shown to impact the transmission and prevention of schistosomiasis [25]. People's level of knowledge about schistosomiasis, as well as water, sanitation, and hygiene (WASH) practices, are closely linked to schistosome infection. For instance, in South Africa, it was reported that caregivers lacked sufficient knowledge, had negative attitudes and poor preventive practices regarding schistosomiasis and soil-transmitted helminths, despite high awareness of the disease [26]. A person's knowledge can affect their attitudes and beliefs, which in turn can impact their practices and increase their risk of infection [27,28]. Similarly, adults' knowledge and beliefs will influence how children are taught and told to behave and act, for example, to avoid playing and swimming in lakes and not defecate in the open. However, changing WASH practices requires much more than raising knowledge levels – at the minimum, adequate water and sanitation must be available for daily needs. Transmission levels of the disease are likely to increase or remain unchanged due to a combination of factors, including the inability to modify high-risk behaviours and a limited understanding of how the disease is transmitted [29].

Several studies in SSA and also in Uganda have reported on the existence of local explanatory models surrounding schistosomiasis – often labelled as "misinformation" and "misconceptions" [3,28,30,31]. A lack of knowledge about the transmission and prevention of schistosomiasis among adults as well as pupils aged ten to fourteen reduced the acceptance of MDA in Uganda [15,32,33]. Moreover, lacking knowledge about MDA or fears around the treatment and its side effects were reasons individuals cited for not taking praziquantel in Uganda [16,31,34,35]. A study in Nebbi (now Pakwach) district at Lake Albert, Uganda, has shown that local understandings of neglected tropical diseases (NTDs) diverged from biomedical understandings. This – in combination with inadequate health education and structural factors – were the identified reasons why individuals did not take praziquantel and other deworming drugs [34]. In some cases, receiving inconsistent information from different sources can lead to confusion and even reluctance to accept treatment or preventive medicine [36].

The relationship between health information and wider contextual factors for the prevalence of schistosomiasis was described by researchers studying *Schistosoma mansoni* in Mwanza, Lake Victoria, Tanzania [37]. According to these findings, schistosomiasis was not considered a priority health problem by adults due to its chronic nature, lack of public awareness of the disease, and poverty in both hotspot and control villages. Yet a lack of leadership, social engagement, community participation, motivation, and commitment to schistosomiasis control were found to be relevant factors associated with persistence of schistosomiasis. The authors concluded that prevalence of schistosomiasis was influenced by socio-political and economic forces that operate across time and sometimes even on a world-wide level, ultimately affecting individuals and communities at the village level [37]. Corresponding studies have not yet been carried out in Uganda.

In preparation for the (pilot) roll-out of the novel paediatric praziquantel formulation for PSAC within the ADOPT-programme, which started in March 2025 [38], we conducted a qualitative study to support the adoption of arpraziquantel in routine health care for PSAC in Côte d'Ivoire, Kenya [39,40], and Uganda. This article explores crucial aspects that will inform effective roll-out preparation by including target populations' awareness of the disease in PSAC and existing knowledge on symptoms, transmission and prevention. Furthermore, this qualitative study aims to identify existing sources of health information that the target population has access to and their preference on how to receive information about the

new treatment option for PSAC. Moreover, the article describes relevant socio-cultural and structural factors as potential barriers to accessing and receiving information in two high prevalence areas in Uganda. By investigating information sources, socio-cultural and structural barriers, and their potential impact on the information flow within the sensitisation campaign, our work provides significant insights.

## 2. Materials and methods

### 2.1. Ethics statement

Ethical approval was obtained from the Ethics Committee at the Technical University of Munich (TUM) (731/21 S), by the Makerere University (MU) School of Social Sciences Research Ethics Committee (MAKSSREC 01.22.528), and by the Uganda National Council for Science and Technology (SS1269ES). All study participants took part voluntarily after reviewing a study information sheet and being verbally informed about the rationale for the study, the study procedures, their rights as participants and being given the opportunity to ask for clarifications. Written informed consent was sought from each participant and an impartial witness in case of illiteracy of the participant. Language barriers were addressed through local research assistants, using translated consent forms. Participants were assured of the confidentiality of the data they provided. The study was registered at clinicaltrials.gov (ID NCT05350462).

### 2.2. Study area and setting

The study was conducted in two districts: Bugiri at Lake Victoria in southern Uganda, close to the Kenyan border, counting a population of 516,735 in 2014 [41], and Hoima at Lake Albert, western Uganda, close to the border with the Democratic Republic of Congo (DRC) with a projected population of 374,500 in 2020 [42]. Through guidance from the Vector Control Division from the Ministry of Health, districts and sub-counties were selected that have a high schistosomiasis prevalence. The combined population of the chosen two sub-counties is approximately 34,000 with 7,500 households.

In Uganda, Schistosoma *mansoni* is the predominant species responsible for causing intestinal schistosomiasis [13]. In these two districts, the overall prevalence of schistosomiasis is high at over 30%, and both sites have a history of MDA with praziquantel, conducted by the government to control schistosomiasis. Besides the distribution of praziquantel at schools, the remaining population (older than five years) receives the medication from health professionals or community health workers, either at fixed points in their village or through a door-to-door approach directly at their homes.

In Bugiri, schistosomiasis prevalence in children between one and five years was measured at 35% in 2014 [12]. A study at the shoreline of Lake Albert reported a prevalence of 46.5% among children aged five to ten years in 2016 [43].

In Bugiri, five villages were selected along the landing site in Wakawaka Parish in Buliidha Subcounty, a fishing community, located on the shores of Lake Victoria. In Hoima, ten villages in the Buseruka sub-county were selected, all located in the Rift Valley close to Lake Albert with a huge escarpment separating them from the more urbanised centres (Fig 1). Due to their geographic location, access to social services is challenging.

In both sites, residents rely mostly on the lake for daily water needs. However, the villages are heterogeneous in terms of water supply: some have boreholes, while others lack any safe water source. In Bugiri, some boreholes were reported to taste salty or as being murky and not suitable for human consumption (Fig 2). There, people have started boiling drinking water on top of the aqua-tab-treated water. Generally, people resort to using lake water when boreholes are over-frequented or distant. Borehole water is mainly used for drinking and cooking, and lake water serves other purposes like hygiene and laundry. In the whole study region, few households have latrines; some rely on neighbours' or community latrines. As not everyone has access to a latrine, many continue practising open defecation.

In the Bugiri region, we observed farming activities and a few small-scale industries, indicating employment opportunities besides fishing. Economically, the Hoima population heavily relies on fishing, but implementing stricter fishing regulations has decreased profits. Generating sources of income other than fishing is challenging as trading centres are far away, and the soil is not suitable for (subsistence) farming. The villages are not connected to the power supply network.

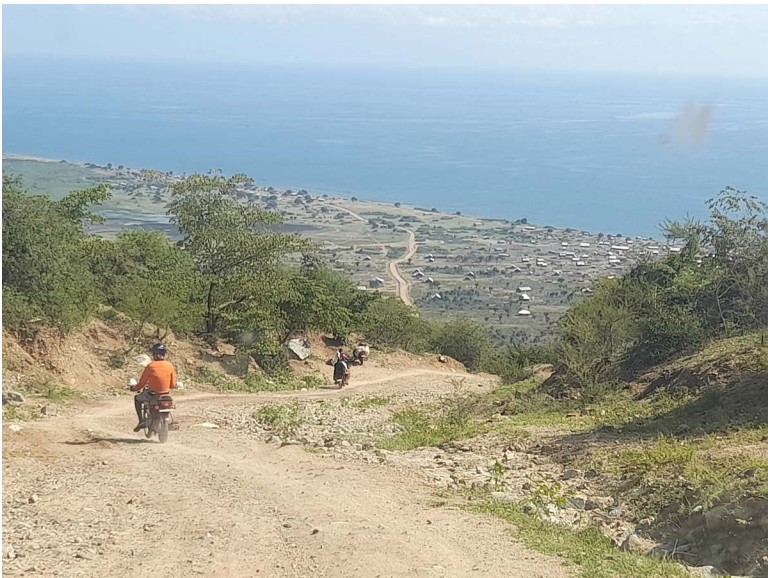

**Fig 1. The steep and rough road to Hoima district at Lake Albert.**

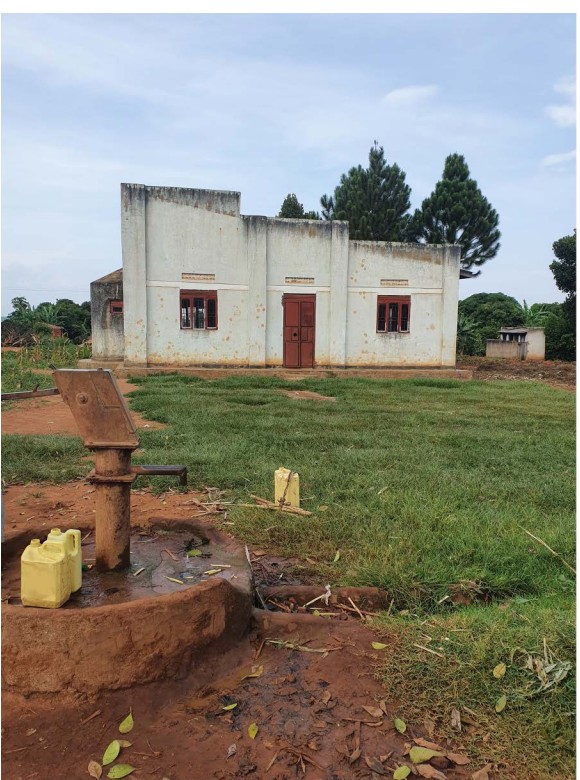

**Fig 2. Borehole in Wakawaka that is described by the community as providing salty water and not used.**

Some small-scale businesses and a few private people use solar panels (Fig 3). In addition, the region is prone to flooding. All roads are unpaved, and some areas are only accessible by motorbike, resulting in further isolation.

In both study sites, there are few public primary schools, some private schools, no nearby secondary schools, and illiteracy among adults is common. During the COVID-19 pandemic, schools in Uganda were closed for two years, re-opening in January 2022, though many students did not return. Proximity to the lake also impacts school drop-out rates: children start earning money by fishing instead of attending school.

### 2.3. Study design, population and recruitment

This study utilised a cross-sectional qualitative design to assess parents' knowledge, preferred information sources and socio-cultural and structural barriers for communicating schistosomiasis information. This enabled researchers to identify and study various factors that potentially affect the implementation of paediatric treatment and to compare these findings across research sites. Researchers conducted interviews with parents/guardians of PSAC; district level authorities, including program coordinators, schistosomiasis program managers and district health officers; community leaders, including the Local Council, opinion/ religious leaders, traditional healers and teachers; health professionals; and community health workers called Village Health Teams (VHT) (Table 1). A total sample size of 143 individuals, comprising of 72 from Bugiri and 71 from Hoima districts, participated in the study. A purposive sampling technique was utilised to select both the villages and the participants for the study. One key selection criterion was that the villages had to be located near the lake shores. We chose interviewees who we believed could provide rich, detailed insights relevant to our research – focusing on individuals with pertinent experience or knowledge. For VHTs and community leaders to meet inclusion criteria, they had to have been long-time residents of the village and active as far as their roles and responsibilities in the community or health system were concerned. Additionally, only those willing to participate were considered.

With assistance from the local council officials or designated guides, our research team visited the households to conduct qualitative interviews. The technique enabled us to select participants based on the groups targeted by the new paediatric praziquantel formulation, including its social mobilisation, sensitisation, and distribution [44]. The selection of the parents/guardians (later only referred to as "parents") was based on the age of their youngest child (below five years,

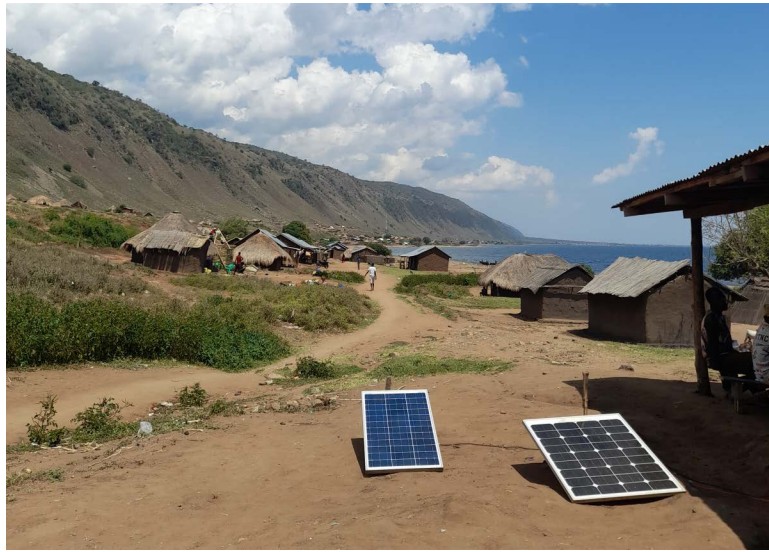

**Fig 3. Missing electricity grid and dependency on solar energy in Hoima.**

**Table 1. Number of interviews.**

| Type of method | Type of participant | Bugiri | Hoima | Total |
|---|---|---|---|---|
| IDI | Parents/guardians | 10 | 9 | 19 |
| IDI | District-level authorities | 3 | 5 | 8 |
| IDI | Community leaders | 10 | 10 | 20 |
| IDI | Health professionals | 4 | 6 | 10 |
| IDI | Village Health Team (VHT) | 5 | 3 | 8 |
| FGD | Parents/guardians | 4<br>(32 individuals) | 4<br>(32 individuals) | 8<br>(64 individuals) |
| FGD | VHTs | 1 (8 individuals) | 1 (6 individuals) | 2 (14 individuals) |
| Total number of interviews | All | 37 (72 individuals) | 38 (71 individuals) | 75 (143 individuals) |

PSAC), sex, and place of residence. The research team leader asked VHTs to identify parents/caregivers who had experience with bilharzia infection and/or treatment for inclusion in interviews and FGDs. Care was taken to include a diversity of profiles (age, gender, education, number of children) in the selection of participants. The sample size was determined by the participants' profiles and data saturation – the point at which the inclusion of new data no longer delivers new information or themes [45].

## 2.4. Data collection methods and instruments

Data collection methods consisted of in-depth interviews (IDIs), focus group discussions (FGDs), and direct observations. Guides for the IDIs and FGDs were employed to collect the data. In total, 19 IDIs were carried out with parents of PSAC. Furthermore, 46 IDIs were conducted with district-level authorities, community leaders, health professionals, and VHTs. Ten FGDs were conducted: four with parents of PSAC and one with VHTs in each of the two districts (Table 1). Given the cultural contexts, the FGDs with parents were disaggregated by sex to allow for freer discussion. Data were gathered through a collaborative effort between a team of researchers from MU and TUM. We recruited two experienced research teams of eight interviewers and trained them in this study's research ethics and general data procedures, emphasising the importance of maintaining confidentiality and informed consent. Fieldwork started immediately after the training, with pre-tests of the research tools to check for logical flow, comprehensibility and translation in local languages conducted preceding data collection. The tools were adjusted, and data collection started immediately after the finalisation of the tools.

The research teams were supported by VHTs, who helped identify households with PSAC. All participants gave informed consent before being interviewed. The IDIs (see supplement S1 Text for the interview guide targeting parents) and FGDs were audio recorded while taking detailed notes. FGDs were conducted with one moderator and one note-taker. Community-level stakeholders, including parents of PSAC, were interviewed regarding their knowledge of the disease, its treatment, and their information sources. The district-level interviews were conducted in English, while most of the other interviews were conducted in the local languages, Lusoga in Bugiri, and Alur and Runyoro in Hoima. All interviews were conducted in private settings chosen by research participants, including homes, schools, health centres, private clinics, and offices. The interviews resulted in 97h24min of audio-recording, with an average time of 1h13min for individual interviews and 1h54min for FGDs.

The researchers carefully documented the communities' infrastructure, looking at access to clean water, usage of lake water, the availability of latrines, the condition of schools and road infrastructure, as well as the electricity grid and access to radio and television, among other aspects. Both research teams held de-briefing meetings every evening.

## 2.5. Data analysis

Interviews were transcribed and translated directly to English, quality checked, and, where necessary, edited. Edits included, for example, correcting grammar to improve clarity and explaining locally used terms. Data were shared via a secure server between the MU and TUM teams and then uploaded into MAXQDA qualitative data management software for organisation and subsequent coding. Based on the research questions and the experiences during fieldwork, the researchers developed a thematic coding tree to analyse the qualitative data. This tree was elaborated and modified based on the triple coding of three interview transcripts and the double coding of two others. LR and MF undertook the coding and discussed with each other throughout the coding process, adjusting the tree when both agreed. Topics and sub-themes, such as quality of information sources, the flooding in Hoima and the stricter fishing regulations iteratively arose during data collection, coding and further analysis, however, reading the scientific literature helped place the topics in relation to one another and review our data accordingly. The researchers added memos and annotations to the software as they coded the data. After reading relevant codes, they were grouped into themes and analysed. Care was taken to identify patterns and differences between participant profiles, including personal demographic characteristics, district location and informant groups.

The analysis was discussed between the TUM and MU researchers. Later, validation meetings were held in each district to disseminate study findings and gather feedback. The feedback that was received was used to refine the results and analysis.

## 3. Results

This section starts with descriptions of the characteristics of the participants, followed by the assessment of existing schistosomiasis awareness and knowledge. It ends with the analysis of interplaying socio-cultural and structural barriers that influence parents' knowledge about, access to, and understanding of information about schistosomiasis in PSAC.

### 3.1. Characteristics of the study participants

We interviewed 19 parents, eight district-level authorities, 21 community leaders, ten health professionals, eight VHTs in IDIs, as well as 64 parents in eight FGDs and 14 VHTs in two FGDs. Half of the parental participants were female (51%). The age of the interviewed parents ranged from 20 to 67 years, all having at least one PSAC in their household. In general, parents from Hoima reported higher educational attainment (many with secondary education and one individual each with a certificate and degree) than the participating parents from Bugiri, the majority of whom reported few years of primary education, but even in Hoima a few had no education. In Bugiri, most of the parents interviewed were farmers, while in Hoima more parents earned their living through fishing or small scale businesses. However, significantly more parents in Hoima stated being unemployed than in Bugiri.

Two of the eight participants from the district-level authorities were female, all with tertiary education or higher. Except for two teachers, all participants among the group of 21 community leaders were male. This group included six chairpersons (Local Councils), eight teachers, two religious leaders, one cultural leader, one keeper of a drug store, one traditional healer, one community opinion leader, and one representative of the fishermen. The age of the interviewed VHTs ranged from 24 to 67 years. They had been serving in their roles as VHTs between four and 25 years.

### 3.2. Existing awareness and knowledge of schistosomiasis

The parental participants in both communities knew that schistosomiasis was prevalent in their communities. Several parents interviewed considered it as serious and deadly if untreated. Generally, parents believed that it affects their lives, although some have grown accustomed to it. Views differed on whether PSAC could carry schistosomes. Parental knowledge concerning signs and symptoms, the transmission, and preventive measures of schistosomiasis diverged from biomedical knowledge.

**3.2.1. Local names for schistosomiasis.** The disease is commonly known as schistosomiasis and bilharzia in the two study areas. However, local names are also used to describe it, as reflected in the following quote.

*Actually, here, since there are many languages, it is "ukoko, ukoko". That is for the language they call Alur, the Alur language. Now for this one [Runyoro, the language] of the natives of this area, I do not understand it well. – IDI, health professional, Hoima (401)*

Table 2 states the names that our respondents reported during IDIs and FGDs.

**3.2.2. Parents' knowledge about the signs/symptoms of schistosomiasis in children.** During the qualitative interviews, the most frequently reported symptoms by parents were swollen bellies, stomach-ache, loose stool/ diarrhoea and vomiting/vomiting blood. With increasing age of the respondents, they reported more symptoms than compared to younger parents. Overall, parents of PSAC knew the disease and could name a few signs and symptoms.

**3.2.3. Parents' knowledge regarding schistosomiasis transmission modes in children.** Generally, parents in both study sites associated schistosomiasis with water, mostly with contact with lake water, swampy areas, stagnant water, or "dirty" water. A few mothers were not aware that exposure to contaminated water is a transmission route. A 48-year-old mother from Bugiri who earned her income from charcoal burning argued that one could only really know about transmission methods after being infected:

*(Laughing) I really don't know. Because I have not suffered from it. You can only know when you have suffered from it. Some people say it is caused by witchcraft and others say it comes from water – water that we drink and is not clean. – IDI, mother, Bugiri (501)*

Few parents were aware that worms are part of the transmission cycle, and amongst those, their knowledge remained limited to simply knowing that worms are part of the process. Very few parents indicated directly that they knew that transmission could be caused by open defecation, which subsequently contaminates fresh water sources. Among these few exceptions, one mother said:

**Table 2. Local names for schistosomiasis.**

| District | Local names for schistosomiasis | Language | Meaning |
|---|---|---|---|
| Both districts | Bilharzia | All | Equivalent to schistosomiasis |
| Bugiri | Mtumbi | Luganda | Abdominal swelling |
| | Entumbi | Lusamya | Abdominal swelling |
| | Kidaada | Lusoga, Ateso and Samia | Swollen belly after being bewitched, often through poisoning the food |
| | Ekidada | Ateso | Equivalent to kidaada |
| | Mpagama | Lugweere | Type of fetish or charm superstitiously believed to have been planted in somebody by an enemy |
| | Bide | Lugweere | Bells |
| | Kabengo | Luganda | Hard feeling in the stomach or side |
| Hoima | Blazia | Runyoro | Bilharzia |
| | Blaja | Alur | Bilharzia |
| | Empuuka | Runyoro | Growing belly |
| | Ukoko | Alur | Diseases caused by worms |

*You see, when people go into the water, they find those parasites that can give you bilharzia. Others drink non-boiled water, then they later complain about stomach-aches, yet they drink dirty water. That is a problem. Some people defecate in the water while they are fishing, and other people come and fetch that same water. All this contaminates the water. – IDI, mother, Hoima (209)*

This quote implies knowledge about open defecation and its relation to schistosomiasis, even though schistosomiasis is not transmitted through ingesting contaminated water. A father cited a lack of hygiene as a risk factor for infection:

*You find bilharzia in dirty places. When you go to places where people defecate and urinate outside the latrines and where runoff water stagnates – you can get bilharzia. – FGD, father, Bugiri (002)*

In general, "dirty waters" might also be a synonym for water where some community members openly defecate. Practices of open defecation were mostly associated with fishermen and children. A few parents associated schistosomiasis transmission with latrines. They described using dirty latrines or walking barefoot into latrines as possible routes for transmission.

Parents that knew that infection could occur in the lake had different theories about how the cercariae (the free-swimming larval stage of the parasite, referred to locally as "germs") enter the body through the water, such as through passing gas, or through the anus, just above the chest, or the feet. Parents identified various other potential routes of transmission, including contaminated food, roasted fish infested with worms, half-cooked fish and soil ingestion, while also mentioning the possibility of disease transmission through flies transferring germs from faeces to food. When asked about the causes of bilharzia and how one gets infected, a 34-year-old fisherman and father of four children between five and 16 described this as follows:

*R6: I may add that bilharzia loves soft skin. It, therefore, enters the children easily because since the lake is our garden, you find our children also there gassing and playing in the lake. And also, the disease enters their skin easily because of the long period they spend playing in the water; therefore, when the disease enters the children, they grow up with it. Secondly, some of us eat fish that is roasted and not dried thoroughly, unlike you who like thoroughly dried roasted fish – for us, we prefer it half roasted. Therefore, you find that these half-roasted fish have the worm for bilharzia in them, and later on, you find someone having a swollen stomach. In conclusion, this is the same water that we drink that gives us bilharzia. – FGD, fathers, Hoima (008)*

The assumption that drinking contaminated water from the lakes causes schistosomiasis was common among parents. Between one-third and half of parents interviewed in Bugiri believed that the disease "kidaada" is caused by witchcraft. Kidaada is perceived as a disease with the same signs and symptoms as schistosomiasis and the term is also used to refer to schistosomiasis. In this belief, a person can bewitch another person to be affected by kidaada by cursing or poisoning their food – mostly said to be done by neighbours. In the following quote, fathers from Bugiri discuss the concepts of kidaada and schistosomiasis (bilharzia):

*R6: Bilharzia is bilharzia and kidaada is kidaada. They are different. Bilharzia is got from water and kidaada is from 'given' food.*

*R1: In Busoga here, there are people with bad hearts who mix herbs that make someone's stomach swell. We have buried two people who died from that type of kidaada.*

*I: Not bilharzia?*

*R1: In Busoga we used to call it kidaada. We used not to separate kidaada from bilharzia. They don't have any difference in signs and symptoms. – FGD, fathers, Bugiri (2)*

As these sentiments show, there are varying beliefs on whether kidaada and schistosomiasis are the same disease with the same origins, or whether they can be distinguished, even though they have different causes. A mother said she used to believe in witchcraft being a cause of schistosomiasis or kidaada, but is now doubting her belief after being educated by drug distributors:

*I hear that they acquire it through the water because those who come to give us the drug tell us that it is spread through the lake, dirty water, especially the running water. Then I used to hear in the past that this disease is also just given to people or one is bewitched, so I am still confused, and I do not know the truth. They used to say that someone can just give it to you, they used to say that someone can put the disease in your food. – IDI, mother, Bugiri (504)*

**3.2.4. Parents' knowledge of preventive measures.** The majority of FGD participants stated avoiding water-logged areas, not staying long in the lake, wearing gumboots, and boiling water as preventive measures – however, boiling water was mostly mentioned in association with drinking water rather than using water for hygiene and laundry. Just as with disease transmission, there were also different ideas about infection prevention. For example, a father and bar-owner with an extended network from Bugiri, responded that one is only at risk of infection if submerged in lake water up to chest-level. Some parents from Bugiri, who believed that witchcraft causes schistosomiasis/kidaada consequently stated other measures to prevent the disease:

*By not eating at other people's homes. By being friendly with neighbours and avoiding creating enemies. By not drinking water at other people's homes. – IDI, mother, 19 years old, Bugiri (505)*

The young mother also named drinking safe water and eating clean food as ways to prevent schistosomiasis. Many parents mentioned proper hygiene, referring to the use of latrines and hand washing afterwards. Several of them referred to praziquantel as a preventive measure.

Most parents, especially those from hard-to-reach areas in Hoima, were aware of their infection risks due to their lack of access to safe water sources. Several parents' voices asked the government for help in providing safe water and sanitation facilities.

*We need to be provided with clean water and enough drugs, and then we shall be able to fight the disease. The main issue is getting clean water and enough drugs, and those are the only ways I know bilharzia can be fought. This is because we drink contaminated water, and we always depend on dirty water.*

*We also need help from the government because, at the community level we cannot manage bilharzia. At the local level, it is very hard to fight bilharzia, and we do not do anything, and this is because we do not have any other source of water; even the water mission pumps which used to provide clean water, the machines went faulty, so we just drink direct water from the lake.*

*They always tell us that bilharzia is got from the lake and swamp water, but we just drink because we do not have any other options. We use the water for drinking and also for home use, so we cannot do much. – IDI, father, Bugiri (503)*

**3.2.5. Parents' knowledge about praziquantel for schistosomiasis treatment and mass drug administration.** Overall, parents had a positive perception of the benefits of praziquantel for their own health and that of their community, including children, even when referring to its side effects. Many parents told us how someone suffering from schistosomiasis had been cured with praziquantel, and from interviews with VHTs and community leaders, it was clear that, generally speaking, the community was grateful for receiving praziquantel during MDAs.

*R3: (..) I cannot refuse to take the drug since I know the benefits.*

*I: Are there your friends, neighbours, or family members who, when they decide not to take the drug, are excluded from public events, or are they stigmatised?*

*R5: We try to educate them on the benefits of the drugs since there are those with little knowledge and, hence, may not understand what the drug is about. So, we persuade them to do the right thing, which is going for the drug. And what is bad, we also refuse as a community. – FGD, fathers, Hoima (007)*

In the IDIs and FGDs with parents, only four participants explicitly mentioned not taking praziquantel themselves: one of them was a mother living with two children in a very hard-to-reach area:

*R: I should be honest to you; I didn't swallow; I didn't pick it.*

*I: Why?*

*R: I didn't get it*

*I: Why?*

*R: (Hhhmm) For no reason.*

*I: This is exactly what I want to know, why don't some people want to get the medicine? Why didn't you get the medicine?*

*R: Hmm… [silence for some time, even after repeating the question several times]. Forgive me. – IDI, mother, Hoima (205)*

The last quote indicates that some respondents did not feel comfortable talking about not taking the medication. Another father of seven children, who chose not to allow his children to take praziquantel, shared during a FGD his reasoning resulting from a previous negative experience:

*R8: An example is me, I also don't allow my children to take drugs anyhow because I had a bad experience. It was around 2017 when my child was given praziquantel at the age of 7 years. The drug treated my child terribly in that I had to spend a lot of money to treat the child. If I had a choice, I would not allow them to take that drug because I end up spending a lot of money treating the side effects. My child collapsed under the sun after being given the drug. When I took my child to the hospital in [location X] we were referred to Hoima referral hospital. Upon reaching the hospital I was told that my child was given an overdose. I spent 450,000 shillings to treat my child. That is why I can never allow my child to take drugs again. – FGD, father, Hoima (008)*

Decisions surrounding reluctance to treat their children with praziquantel are born out of personal experiences. Parents often told us about others in the community who would not take praziquantel.

*R3: Honestly, I cannot guarantee complete acceptance because you can find one parent refusing and another one accepting his child to participate. – FGD, mothers, Bugiri (003)*

*R7: Some parents are just stubborn. They think that since they don't have signs of the disease then they are fine, and therefore they think there is no need to take the drug. – FGD, fathers, Hoima (008)*

Health professionals also told us about sick community members who looked for alternatives to praziquantel. Some of them resorted to home remedies for schistosomiasis or sought help from witch doctors or traditional healers.

In general, the community recognised praziquantel as a positive control measure and treatment drug, largely appreciating its positive impacts. Community members with differing opinions may not always have opened up about them in interviews, but data triangulation shows that some community members favour alternative treatments.

### 3.3. Available, trusted and desired information sources about schistosomiasis in children

After identifying local ideas and explanations among parents on transmission routes and preventive measures, that diverged from biomedical explanations, we subsequently studied how parents received information about the diseases, especially among children and asked through which channels they would like to be informed about the new paediatric praziquantel. In addition, we triangulated the responses by all informant profiles to verify parents' responses and to evaluate the quality of the information currently provided to parents (see Table 3).

The preferred sources of information about the MDA with the new paediatric praziquantel included VHTs, health professionals, the Local Council, and, in some sites, the radio. The data revealed a parental preference for VHTs to work in conjunction with health professionals. Radio was suggested more often in Bugiri than in Hoima, though there were concerns about its effectiveness due to limited radio ownership and reception issues in both sites. Our study identified challenges of using text messages or mobile apps due to limited smartphone access, emphasising the importance of peer-to-peer exchanges and community involvement. Parents in these areas would have preferred verbal communication through radio, TV, or community leaders, but the availability of TV and radio posed challenges.

**3.3.1. Village health teams as information sources.** We explicitly inquired about trusted sources of healthcare information in the parent IDIs. VHTs tended to be cited more frequently as trusted messengers by parents in Hoima than in Bugiri, especially by parents living in remote areas, as a mother in the following quote explains:

*R: In this area, we get communication from the VHTs. Sometimes, nurses come from government hospitals and administer drugs to our children, like in the case of polio.*

**Table 3. Summary of findings: Enablers and barriers to schistosomiasis-related information.**

| Theme | Sub-themes | | | |
|---|---|---|---|---|
| **Available and trusted information sources & parents' preferred sources for arpraziquantel** | Village Health Teams (VHTs): trusted, but parents have doubts about their education | Health professionals: trusted but in remote areas not available | Village leadership: trusted and parents are accustomed to, but more as mobilisers than as informers | Radio: sometimes not available due to the remoteness of the region and no access to the power grid, especially in Hoima |
| **Quality and content of information, provided to parents** | Quality of information provided dependent on knowledge of the parents' information source: lower information quality provided by VHTs and community leaders, higher quality provided by teachers and health professionals | | | |
| | Information content provided to parents is sometimes limited to the time and place of the drug distribution | | | |

| **Socio-cultural and structural barriers to schistosomiasis communication** | Language barriers make some communication difficult: this applies mostly to teachers and health professionals but also to people who migrated, such as refugees | Low literacy rates reduce information sources to verbal ones | Population movement: people miss out on information | Fears concerning praziquantel | (Impact of) Stricter fishing regulations resulted in mistrust in government programs | Mistrust in health campaigns (post-COVID19) |
|---|---|---|---|---|---|---|
| **Consequences & strategies to overcome them** | Communication challenges can affect understanding. Use verbal communication complemented by pictures; use translators | | Danger of missing out the target population. Repeat sensitisation, collaborate with key players such as the fishermen's association | Rumours about the "real" aim of the treatment and fears concerning praziquantel can affect acceptance. Consider rumours and fears during the development of sensitisation campaigns | | |

*I: Why do you trust such sources?*

*R: Because we lack other effective sources, for example, our frequency waves for radios are poor, as is our phone network. – IDI, mother, Hoima (207)*

VHTs themselves described their role as disseminating health-related information within the community. Their primary focus was on logistical details, such as informing the community about the timing and location of vaccinations and treatments (mobilisation). According to interviews with VHTs in both sites, they also educated the community on preventive measures and promoting proper hygiene practices at home. VHTs also referred community members to health centres for treatment when needed.

Parents in both Bugiri and Hoima stated a preference for VHTs to inform them about paediatric praziquantel. However, some participants, especially from Bugiri, expressed concerns that VHTs alone might not be sufficient due to their lower levels of education compared to health professionals.

*From the facility, yeah, because those people are professionally taught about the drug. VHTs can also pass the information because they are trained, and they can pass it to the community, but the most trusted are the doctors. – IDI, father, Hoima (206)*

*R1: As the drugs come, find health workers who can ably explain to the parents what the drug is going to do in the 1–5 year old, someone who can answer questions very well – not the VHTs who people know that they didn't go to school. The program should use qualified health workers who will sensitize on the benefits of taking the drug and the possible side effects. – FGD, fathers, Bugiri (002)*

**3.3.2. Health professionals as information sources.** Proximity to health facilities influenced the preference for health professionals. In Bugiri, parents indicated a preference for consulting health professionals and receiving information from the radio when asked about their trusted sources of health care information.

*R4: Health workers from the government health facility should be the ones to sensitise on the new drug. They are professionals in this.*

*I: How should they do it?*

*R5: Through the Local Councils and the community announcer. They should make announcements about the distribution of the new drug. They should specify the central places where the community members should gather.*

*I: Why not VHTs?*

*R1: VHTs are not listened to because community members know they are ordinary people from villages who can't explain much. Health workers from the health facility will be listened to because they will have put in a lot (of time) to come and talk to the community. – FGD, fathers, Bugiri (002)*

The services at the only Health Centre III (higher level health centre), serving the study villages in Hoima, were disrupted due to the 2020 floods. A health professional described the challenges experienced in the following quote:

*(…) water flooding, and the challenges of accessing health and health services became a priority challenge because it took us time to shift from there [the previously flooded health centre]. (…) Then, every person had to go on the boat to go there, which was hampering access to health, but after three months, we shifted, and we actually have these temporary tents as you see now. – IDI, health professional, Hoima (401)*

Moreover, the community members expressed dissatisfaction with the temporary relocation of the health centre to tents (Fig 4), citing concerns such as the lack of privacy.

**3.3.3. Community leaders as information sources.** In addition, we examined the role of community leaders, including political, religious, and traditional leaders and teachers, in providing health information. The Local Councils (also L.C., including its Chairmen), representing the village leadership, held positions of high respect in all villages, and parents noted that they valued their guidance. When Local Councils called for meetings on health campaigns, VHTs and professionals from the health centre provided more comprehensive explanations of the relevant issues. The Local Council emerged as a trusted channel to facilitate the information flow about the campaign for PSAC:

*R6: To make our colleagues agree, they should have general meetings for parents so that everyone attends, and this should be done through the Local Council. They can even use the loudspeakers to mobilise and also welcome the presence of the partners in this new invention, this should not be left to the VHTs alone. – FGD, fathers, Hoima (008)*

A popular sensitisation method was for someone from the Local Council's administration to move around the neighbourhood with a megaphone or for loudspeakers to be set up to broadcast messages that inform the public.

Because teachers are actively involved in the MDAs, i.e., they are present when pupils experience the effects of the drug, they play — or at least could play — an important role in educating children about preventive measures. Teachers who are in close proximity to children during school-based MDAs told us that they inform children at school about MDAs with the idea that children then pass this information on to their parents.

## 3.4. Quality and content of information provided to parents

**3.4.1. Knowledge among providers of information to parents.** Among VHTs and other community leaders, knowledge about signs and symptoms was similar to that of parents, with a high variance of knowledge regarding schistosomiasis symptoms among VHTs. Two individuals from the group of community leaders, a religious healer and one

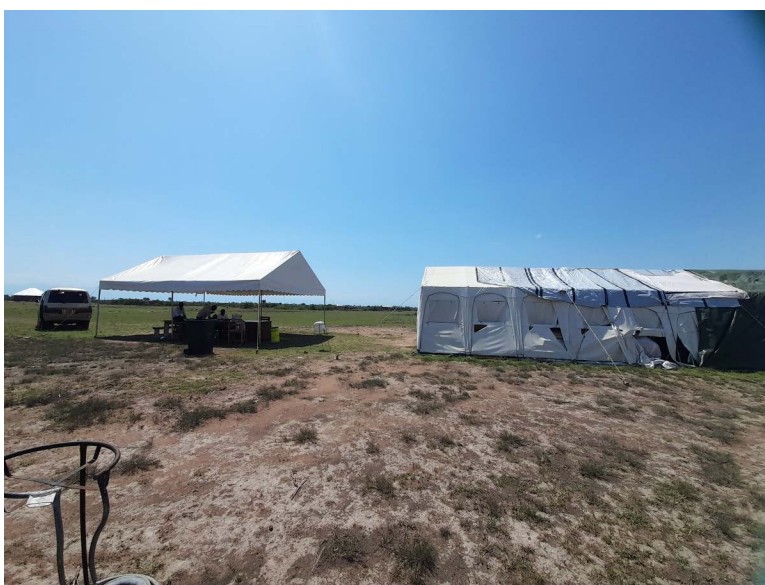

**Fig 4. Temporary tented Health Centre III in Hoima.**

teacher (who had moved recently from a different region), could not name any symptoms. In contrast, seven out of the eight teachers were well-informed about signs and symptoms.

Knowledge about schistosomiasis transmission and prevention, among VHTs and community leaders sometimes contradicted Ministry of Health information materials based on biomedical knowledge. Some messages conveyed questionable information, potentially based on misguided disease understanding, as opined below:

*I always mobilise people to take their drugs. For example, recently, we just gave out the bilharzia drugs, so we first mobilised people in the community to take drugs. We mobilise community members to have latrines, bathrooms and also where to hang utensils after washing in their homes, and all these are found in the category of health. We teach them so that they try to prevent other diseases like bilharzia. This disease of bilharzia is caused when someone openly defecates, and when a fly lands on the stool it carries it and drops it on food and this can spread the disease. – IDI, VHT, Bugiri (612)*

In general, knowledge of disease transmission through water contact was modest. Most VHTs, community leaders and teachers knew that schistosomiasis infections were associated with water and open defecation and identified children as a high-risk group due to water play. However, VHTs provided different explanations for transmission: germs entering the body through wounds, genitalia or flatulence – or mistaken beliefs that schistosomiasis could be caused by flies or eating uncooked food. Few knew that the disease life cycle involves snails, while others knew that snails are involved. It became clear that local explanations for schistosomiasis transmission among the communities – to which the VHTs and community-leading figures belong – were widespread and that these informants also played a role in the extent of parents' knowledge. In contrast, health professionals had adequate knowledge of the transmission route, including the link between open defecation and contaminated water. One health professional in Bugiri referred to the lake as a "toilet of ours", reflecting that it is an often used place for defecation, and is linked to the scarcity or absence of latrines.

The importance of praziquantel in decreasing schistosomiasis transmission was emphasised by VHTs, health professionals and district-level personnel. Still, they also named structural and practice-based factors, such as clean water access and latrine quality, as key issues for disease prevention.

**3.4.2. Content of information provided to parents.** From our interviews with parents, we did not hear about any recent schistosomiasis information campaigns that systematically provided information about symptoms, transmission and prevention. Mothers from Hoima reported during an FGD (009) that VHTs advised them on general hygiene and health. VHTs, health professionals and other community leaders stated that they all informed parents on child health, but this seemed to take place in an unstructured manner on a case-by-case basis with a focus on treatment.

Concerning MDAs, parents were mainly informed by VHTs at community meetings about the logistics (place and time) and were advised to eat before taking the drug to reduce nausea and other side effects. They were also informed that having diarrhoea after taking praziquantel indicated infection clearance. Besides the above, they did not receive information concerning potential side effects and how to treat them.

*(...) They tell us that you are going to take these drugs but when they find the disease in your body you are going to develop diarrhoea. Then they do not talk about these other side effects, but we just experience them from there after swallowing. – IDI, mother, Bugiri (504)*

*Yes, health workers (nurses) tell us about side effects, but they advise us to ignore them arguing that it is the medicine that can be doing its work after finding the disease in the body. – IDI, mother, Hoima (203)*

Concerning MDAs at schools, there appeared to be a communication gap between the schools and parents. While teachers instructed pupils to inform their parents to pack food for the MDA on the next day and community meetings were held, some parents seemed unaware of the upcoming MDAs at school.

### 3.5. Socio-cultural and structural barriers to schistosomiasis communication

Low literacy rates and restricted access to official information sources, relying on the scarce information from VHTs, whom some parents did not trust, and health professionals, further complicated by language barriers and a high population movement, were challenges for accessing schistosomiasis information, sustainable health education, communication and subsequent health literacy.

**3.5.1. Language barriers and high level of illiteracy as communication challenges.** In the study regions, community knowledge levels about schistosomiasis face significant obstacles due to language barriers and a high level of illiteracy. Most health professionals and teachers assigned to the study sites do not originate from the region; they often come from different regions and cultural backgrounds, speaking diverse languages.

*For the case of school, the performance at school has declined. Then we also have some parents who have negative attitudes towards education and whenever you call them for a meeting, they do not turn up. That's for the case of school. Then for the case of the community, most teachers in this community have language barrier challenges. – IDI, teacher, Hoima (21)*

Depending on their previous state of residence, not all had prior exposure to schistosomiasis and are only gaining insights into the daily challenges of the local communities after some time. A few statements, like the one above, indicate that some teachers and health professionals might struggle to establish a close connection with their new communities.

*I: Is there a local term used for that disease?*

*R: Okay, others say "empuuka". (…) It is in Runyoro.*

*I: How about the Alur?*

*R: I don't know; we just get someone to translate for us. Here language is a problem, it is a language barrier. – IDI, health professional, Hoima (405)*

*I: What is the local meaning of kidaada?*

*R: Now some of the local people think it is bewitched. Those are the minds of the people and that is the local language, though I am not fluent. – IDI, health professional, Bugiri (703)*

Several statements from health professionals and teachers highlight that language challenges between health and educational structures and the community exist in the sites visited. Although some local languages are related, communication can still be hindered by not understanding the local context, leading to misunderstandings, especially for sensitive or technical conversations. Furthermore, teachers who do not speak the local languages and teach in English face difficulties in effectively communicating with parents.

Besides the challenges related to different languages, the high rate of illiteracy reduces the number of effective communication channels as text-based material reaches only few community members. In both sites, education is hampered by the low number of schools, a high number of pupils per class, long walking distances to primary schools, and no secondary schools in the study areas.

*The children study, but school fees are a problem. Government schools are very far, and the younger children can't walk that distance. The ones we have here are only two and they are private. You find that sometimes you are not working and you fail to get school fees. (…). They end up staying home (…). Here, there's no government school. – IDI, mother, Hoima (209)*

*R8: Different pieces of advice should be given to such people given the fact the majority of people in this community are illiterate and have never reached primary one; hence, it's of great importance to use the Local Council to educate these people as well as mobilise and sensitise them about the benefits of taking this tablet. – FGD, fathers, Hoima (007)*

Illiteracy and a lack of education were brought up as being major problems for the communities.

**3.5.2. Information interruption due to population movements.** The high fluctuation among the population and the prolonged absence of fishermen over days were apparent in both sites, challenging the effectiveness of health campaigns. Many refugees and migrants, mostly women and their children fleeing from armed conflicts in DRC just at the opposite shore of Lake Albert, start a new life in Hoima. This continuous flow of refugees to the study site in Hoima challenges providing services through VHTs, who typically know their community members well. Newly arriving people are most often not familiar with the existing structures and do not know the responsible VHTs or health centres. Additionally, in Bugiri, families change their residence often, impacting the continuity of praziquantel delivery, as described in the following quote:

*You find that this time they have got treatment here in Wakawaka but next year when it comes to give them another round [of praziquantel], they would have immigrated or shifted to another..., so you find that they move according to how much they are fishing. If it's depleted in Wakawaka, tomorrow, they go to another place where fish are being caught in big numbers. Then we also have migration. There are also these inlands [migrants within Uganda], they come and settle at the lakeside there, but sometimes again they also move away, like in Wakawaka one time we had people who moved to Kiboga district because there is a fertile land. So they had to go, and then we had a depletion of people in Wakawaka. So, these immigrations also cause a non-consistency in treatment. – IDI, district officer, Bugiri (802)*

The 2020 flooding in Hoima not only took a toll on its Health Centre III but also left many homes in ruins. As a result, affected families dispersed to different houses, villages, or regions. We also heard other reasons for the high level of population movement, such as seeking more fertile land. Fisher families were generally described as nomadic: participants explained that families move where they find the most fish, which can mean changing their home every six months.

**3.5.3. Fears of praziquantel and mistrust in governmental health campaigns.** Fears surrounding praziquantel, suspicions about the government's true intention behind health campaigns, negative experiences with government policies, and rejection of the ruling political party were identified as key themes contributing to fear, mistrust, and the undermining of health promotion campaign effectiveness.

During the interviews and discussions, we noted that several fears and rumours concerning schistosomiasis and its treatment were spread among parents in the targeted communities. The rumours ranged from narratives that VHTs and health professionals were stealing schistosomiasis drugs from MDAs to sell them privately, to fears that the drugs would be part of a hidden family planning agenda (or perceived worse scenarios), ultimately harming or even killing children. In Bugiri, we also heard that parents associate strong (side) effects of the drugs with potential overdosing.

*R4: Sometimes most parents do not allow their children to take the drug because the VHTs may, at times, forget to move with the measuring scale. That's why they refuse to give the children the drug because they may think that the children may be given an overdose.*

*R3: Some parents have different perceptions concerning this programme. They think the government has developed a plan to harm their children. Other parents think that the government has planned to stop the mothers or women from giving birth to many children. – FGD fathers, Hoima (008)*

A few parents appear to be generally critical of medical treatment for children. In both sites, parents gave examples involving vaccination campaigns, including recent COVID-19 vaccinations, expressing fears that vaccines were harmful or could even kill people:

*R2: Some people have misconceptions that the vaccines will reduce the life span of their children. They tell their children not to go for vaccination. – FGD, fathers, Bugiri (2)*

*Recently, when they brought the corona medicine, there was a rumour that it kills people. That when you're vaccinated, you will die. – IDI, community leader, Hoima (306)*

In addition to the recent COVID-19 vaccination campaigns, negative sentiments towards government-led initiatives can be traced back to unfavourable experiences with the enforcement of stricter fishing regulations. We have gathered accounts of fishermen being apprehended, with some reportedly still in custody. For those who comply with the new fishing regulations, the reduction in income, combined with inflation, burdens their economic well-being doubly. Residents of Hoima, in particular, have seen a decline in income since the introduction of these policies, while in Bugiri fewer voices mentioned the impact of stricter fishing regulations.

*People have been affected by the fishing activity there [in Hoima]. Now the government is implementing some projects where they want only big fish to be caught. And that now has affected most of the residents as they used to use these substandard nets that are able to give them small fish – which are easily caught compared to the big ones. You may get [only] one [fish]! – IDI, health professional, Hoima (401)*

*There are no schools – both primary and secondary – in this area of [name of village]. Some of our children attend school in Hoima town, and due to the destruction of our boats and nets, we are unable to pay their school fees. – IDI, mother, Hoima (203)*

*…since they started controlling the fishing industry, people lack what to do/are unemployed, yet previously, one could go to the lake and come out with some catch to earn some money to survive, which is not the case today. So, those people who had the general purpose fishing nets are no longer working because all their nets were burnt. So, there is a high level of unemployment. – IDI, community leader, Bugiri (602)*

In Bugiri, residents have more options to earn a living, such as subsistence farming and trading, unlike in Hoima, where infertile soil, low rainfall levels, and villages' remoteness limit such options. We witnessed a sometimes-critical attitude towards campaigns from the government or, in some cases, even a rejection thereof. In Hoima, reports indicated that certain individuals refused to allow VHTs entry into their homes due to a general fear of government officials. This highlights the evident tensions arising from the enforcement of stricter fishing regulations, the consequent economic losses, and the subsequent growing resistance towards government initiatives.

However, the political landscape also plays a role in Bugiri. While the majority of parents in Bugiri stated they accept health campaigns when the government organises them, one health professional mentioned "political imbalances" and explained that some community members preferred to receive information about health campaigns from certain political leaders or parties that they support – and not from others. They recommended using the health structures for community sensitisation rather than higher political leaders (whom some people reject). These findings underscore the influence of fears and rumours, the current political climate, and the ensuing sentiments on shaping perceptions of other government programs, including public health campaigns.

## 4. Discussion

In this research project, we used a qualitative approach to study knowledge and explore access to information on schistosomiasis of parents of PSAC. We also identified related socio-cultural and structural barriers to assessing and receiving health information, such as language barriers, results of population movements and the impact of natural disasters and other government programmes. Our findings suggest that accessing and understanding accurate health information is influenced by multiple socio-cultural and structural factors in our study areas (Fig 5).

### 4.1. Parents' awareness and knowledge of schistosomiasis

In accordance with other studies that were conducted in high-prevalence areas in Uganda [28,46,47], and in line with a review of schistosomiasis knowledge, attitudes, and practices in SSA [3], our findings show that the communities are aware of schistosomiasis and most respondents know the long-term consequences. Despite their awareness, parents' knowledge of schistosomiasis in PSAC varied and we detected modest biomedical knowledge concerning prevention and transmission. This is in line with a study among caregivers of PSAC from South Africa that detected inadequate knowledge, attitudes, and practices despite high awareness of schistosomiasis and soil-transmitted helminths (87% and 79%, respectively) [26]. In our results, we found parents not differentiating between preventive WASH measures for schistosomiasis and other faecal-oral diseases. As in other studies [3,46,48–51], boiling drinking water, for instance, was frequently mentioned as a preventive measure, despite schistosomiasis having a different transmission route.

We also detected other local ideas and explanations regarding schistosomiasis causes and preventive measures. In Bugiri, the belief that someone can bewitch or curse a neighbour and cause them to suffer from a schistosomiasis-like disease called kidaada persisted amongst approximately one-third of the respondents. In line with our results, several researchers in Uganda described communities stating witchcraft as one of the sources of infection [28,29,36,47,52], as well as a corresponding study by the Pediatric Praziquantel Consortium that was conducted in Kenya [39]. If community members have different understandings of transmission, such as the belief in witchcraft, it may hinder treatment efforts. Continuous effort is needed to address the different perceptions of schistosomiasis transmission and to interact with treatment providers in Bugiri. The reality of witchcraft as a meaningful force in the world for a significant proportion of our participants would be important to take into consideration for future sensitisation efforts, possibly through cultural mediators or community dialogues. Using the term "ekidada/kidadaa", the term associated with witchcraft, to refer to schistosomiasis is not recommended, as this approach led to confusion in other settings [36]. The confusion created (schistosomiasis and kidaaa relate to different disease concepts) discouraged involvement in MDA campaigns.

Concerning knowledge about using praziquantel as preventive therapy, we noted that participants sometimes exhibited discomfort when disclosing that they did not take the medicine during MDAs, and some avoided talking about it. A single mother from a hard-to-reach village even apologised to us for not taking the drug. However, this could be explained by the effect of a social desirability bias: As our study team was collaborating with the Ministry of Health, respondents might have provided the answers they thought would be correct and acceptable. Similar findings were reported in another study from Uganda, where people avoided admitting that they did not swallow praziquantel, even if they had collected it [34]. These researchers identified links between deciding not to take the medication and low health education, as well as with rumours and conspiracy theories regarding the true purpose of treatment [34]. These sometimes-guarded answers and

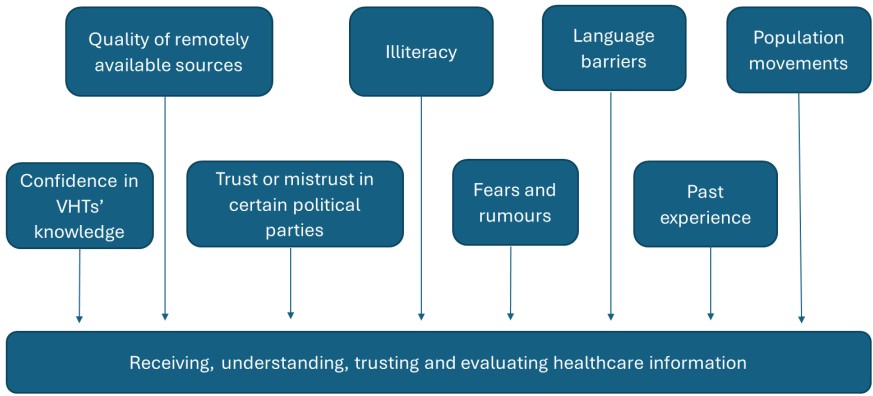

**Fig 5. Socio-cultural and structural factors posing challenges to accessing accurate healthcare information.**

the emotion expressed around participating in praziquantel campaigns indicate a need for a considered approach that recognises the sensitivity of the subject for some members of the community, and to take the time and effort to address their concerns.

We identified fears among participants that health campaigns could be part of a hidden governmental family planning agenda or even plans to reduce the population. Several studies identified fears that praziquantel may affect fertility [34,36,53]. One of these studies found links between local resentment against the government and the concerns about side effects of praziquantel in Panyimour, Nebbi (now Pakwach) district, North-eastern Uganda [34]. Claims were made that government officials intended to reduce the population. They concluded that the lack of information made available for the mass treatment fuelled conspiracy theories [34]. Such findings show the importance of intense sensitisation of the recipient populations before introducing a new health intervention. Trainings for VHTs and community leaders are necessary to enable them to inform parents sufficiently. It is also crucial to address existing fears and rumours effectively by developing targeted strategies to counteract them. Providing more information on the effects of the drug could improve informed decision-making and demonstrate that authorities take community members' fears seriously, thereby getting ahead of rumours that may otherwise circulate.

From 2014 to 2022, MDAs in Bugiri and Hoima achieved coverage of ≥75%. Despite this, the latest recorded schistosomiasis prevalence in 2022 remains moderate (10–49%) in Bugiri but high (>50%) in Hoima [54]. However, endemicity status can vary within districts, making district-level classification an imperfect representation of true community-level prevalence [55]. A key factor contributing to high schistosomiasis prevalence is the lack of access to safe water. Acknowledging this, several respondents asked for help from the government. These outcries for support from "above" indicate that many people know their infection risk but feel helpless about prevention since they do not have access to safe water. The situation concerning toilets and latrines is very similar. In one village, the houses are built so closely together leaving no room for toilets. Some of the latrines they dug themselves are too shallow, resulting in faeces flushed into the lake during heavy rain and floodings. There were communal toilets, but they were unlocked and not very clean – or locked requiring request and retrieval of the key from someone. In view of the unsatisfactory sanitary infrastructure, it is unlikely that open defecation will decrease, despite knowledge about schistosomiasis transmission.

### 4.2. Parents' current information sources on schistosomiasis

Our findings show that parents are informed about schistosomiasis mostly by their VHTs and health professionals. While at Lake Victoria the radio is a relevant information source, at Lake Albert the situation is different; radio is mostly not accessible in the remote villages lying in the Rift Valley. Therefore, community meetings, called by the Local Councils, play a more important role. Our findings confirm the results of an earlier study about PSAC in Uganda, identifying health workers at health facilities and community health workers, radio and community members as caregivers' information sources, but interestingly also the research assistants administering the research survey [52] – as experienced by our research team as well. One respondent claimed that instead of asking questions about schistosomiasis, we should rather come to inform the community about it. These findings indicate a gap in the availability and flow of information. We conclude that better health information infrastructure is needed, with a particular focus on parents and schistosomiasis information.

In contrast, a recent mixed-methods study in Western Uganda analysed a community's primary sources for information on schistosomiasis, assessed through a survey: family, friends and neighbours (28.1%), radio and health workers (both 24.3%), along with mentions of learning about the disease at the lake (6.6%), from teachers (3.8%), and through brochures, posters, and others (3.5%). Some participants, especially those living further from the lake, were unaware of the disease [28]. Our research also revealed that people who moved from less exposed areas, like teachers, lack knowledge about schistosomiasis. The impact of living location can be explained by the fact that people from less prevalent areas have neither experienced nor observed the disease and were not targeted by information campaigns. This also indicates that information campaigns effectively increase knowledge.

Our study also highlights the importance of VHTs and the challenges that they face to fulfil their tasks. Despite our finding that VHTs were parents' primary information source and are trusted because they are part of the community, caregivers often expressed concerns about the limited knowledge of VHTs, leading to mistrust due to their lack of medical training. The Ministry of Health should look into ways to increase VHTs' knowledge through improved and continuous training, including modules on health communication and diverse cultural understandings.

While respondents in our study rarely cited movies as sources of information for schistosomiasis, a Tanzanian study showed that screening this media leads to an increased understanding of the transmission cycle [36]. A common finding of this study [36] and ours is that many respondents learnt about schistosomiasis through both informal channels and through experience, which can be observing friends and neighbours falling ill and associating the symptoms with schistosomiasis. Previous results from Uganda describe the disadvantages of community members informing other community members, namely that key details changed and were influenced by both a poor understanding of the biomedical facts and already entrenched local beliefs about causes and treatment [36].

A study from South Africa showed that caregivers who received health information through community meetings showed better knowledge of schistosomiasis than receiving it from informal channels such as family and friends [26]. This underscores the need for accurate, comprehensive, and clear health information to avoid knowledge discontinuity. Comprehensive information material that assists VHTs and community leaders with providing information to their communities could be helpful in keeping information correct.

### 4.3. Parents' preferred sources of information for the new paediatric treatment

VHTs, health professionals, radio, and the local council were the most preferred sources of information for the new paediatric treatment by parents. Our findings are in line with the results of a recent study from Kagadi and Ntoroko districts along Lake Albert in Uganda, which identified the so-called community radio (megaphones and portable radios/loudspeakers) as the most preferred method to receive information on schistosomiasis. At the same time, VHTs and the village chairman were the preferred persons to deliver such information, also through door-to-door visits [56].

With health professionals and the radio having the disadvantage of rarely being available in remote regions, we identified VHTs and the local council as the most accessible and accepted information distributors. Both require training to increase their knowledge of the disease and to educate them about the new medication, as well as facilitation with easily understandable messages and sensitisation material.

### 4.4. Socio-cultural and structural barriers to health campaigns

A defining socio-cultural characteristic of the two study sites is the diversity of languages spoken by different individuals. While language barriers have been identified as hindering factors in accessing health care in East Africa [57] and worldwide [58] and are a problem that especially migrants are confronted with [59], the topic does not seem to find much attention at the policy level. The results of our study confirm these challenges and indicate that language barriers are a structural problem, as the government posts teachers and health professionals in public employment anywhere in Uganda, irrespective of their language skills. For the provision of information on schistosomiasis, this means that communication may be hampered but moreover, these experts may have a different cultural background and may have had little exposure to schistosomiasis beforehand and thus lack disease-related knowledge. Making use of visual communication material and reaching individuals through a peer-group approach could tackle this challenge.

Our current research also shows that population movements hamper the continuity of health (information) campaigns. Travelling fisherfolk communities, one of the two groups with high infection rates, might not receive announcements signalling upcoming MDAs, thereby missing them. This is in line with the reasoning that human migration plays a major role in the spread of NTDs [60,61] and other diseases; and population movements are expected to increase the schistosomiasis prevalence [25,29,62]. The challenge of movement was already described in an earlier study on schistosomiasis

control among Ugandan fisherfolks [35]. The long absence of fishermen, in combination with a high percentage of migrants and refugees earning a living from fishing, is a double challenge for information campaigns. The stricter fishing regulations could also foster more population movement if policies in neighbouring countries are less restrictive. Educating a population that experiences continuous change or is partially absent can prove tricky, particularly when it comes to disseminating information about preventive measures and keeping them informed about health-related activities. Distributing health information through community radio could help to expand the reach of health campaigns in areas with broadcast coverage.

The impact of natural disasters and government regulations on fishing were identified as structural barriers. During our field visit, residents extensively discussed the devastating flood that occurred in Hoima in 2020, which also affected the only existing Health Centre III. Its service was interrupted for a while and still operated in tents during our data collection. This challenging situation sheds light on the strong attachment of parents in Hoima towards their VHTs. In contrast, parents in Bugiri had more hesitation towards VHTs and were more in favour of receiving information about the new treatment from health professionals. These examples highlight the contextual differences among areas, requiring a sensitive balance between following standardised strategies and messages versus adapting health campaigns to the local context.

We noted suspicions against government programs in our study from certain individuals, especially those referencing the new fishing regulations. Respondents said that they feared the so-called Marafuku, government officers who enforce the new, stricter fishing regulation, and would not open their doors due to the fear of government officials. This example shows that fear and rejection of some political activities can or may also lead to the rejection of other initiatives if they are perceived as political, which could result in rejecting a new health campaign, and how choices made about medication are not made in isolation from wider relations of power and larger political and economic forces. The significance of underlying political and economic situations was also pointed out by other researchers. They concluded from a comparison of different sites in Uganda that the success of NTD control in one district (Busia) was promoted by the positive political and economic situation of this district because the knowledge level of this population was as modest as that of the compared districts, that had less success with MDAs [34]. This finding could imply that government health campaigns are more likely to be accepted and effective in a stable, well-perceived economic and political environment. In cases where there is already a negative attitude towards government programmes, sensitisation could be more successful if carried out by more neutral parties such as health professionals or VHTs.

## 4.5. Limitations in our study

Depending solely on VHTs as guides to identify households with PSAC to participate in the study may have introduced a selection bias, i.e., that certain households could have been either intentionally or unintentionally overlooked, especially households that are not in good contact with VHTs. One parent was recruited independently from VHTs' advice and was one of the few persons who said they did not take praziquantel themselves. At the same time, it was a thoroughly deliberated decision as VHTs are generally trusted locally and were, therefore, helpful in arranging the appointments.

The presence of various languages among participants was addressed by employing local research assistants who each spoke several languages. However, most authors of this article depended on translated transcripts. The co-authors from Uganda reviewed the analysis and interpretation accordingly. Having VHTs and researchers from Germany present during the study and introducing the study as being part of a consortium that includes the Ministry of Health may have potentially influenced responses due to "social desirability" of answers or non-compliance. To avoid such effects, IDIs with parents were only conducted by local researchers, and German researchers also stayed in the background during FGDs with parents. Following the guidance of the experienced Ugandan team, German researchers interviewed informants who were accustomed to having contact with foreigners and were less likely to feel uncomfortable.

Finally, one could observe that while we interviewed both male and female caregivers in equal numbers, the majority of our informants were men. These respondents held positions of authority or function that were determined as having key

insights into schistosomiasis interventions in the focal communities, and may have skewed our analysis towards a more male-centric perspective.

## 5. Conclusion

Preschool-aged children do not currently receive preventive chemotherapy against schistosomiasis, mostly due to the lack of a suitable formulation of praziquantel. As the new formulation arpraziquantel is child-friendly and was tested as safe and effective for children from the age of three months, PSAC could soon also benefit from routine schistosomiasis treatment. Information campaigns that target caregivers of young children require thoughtful design and adaptation to local conditions to overcome socio-cultural and structural barriers. According to our substantial thematic analysis, there seems to be a preference that implementation needs to be carried out by trained and motivated VHTs and, at best, supported by health professionals and through the administrative structures of the local leadership. These are essential components to inform parents of PSAC so that they can make well-informed and trusted decisions for their children's health. While in-depth qualitative research is not intended to be widely generalisable, our findings highlight the importance of considering contextual factors – such as language barriers, level of literacy, population movements, fears, rumours, and trust in government programmes – for planning the (pilot) roll-out of the schistosomiasis treatment for PSAC also in other endemic countries. Any identified challenges should be addressed through targeted interventions, such as visual information material, community dialogues, and community radio to enhance health information flow and build trust. By using contextualised approaches, local barriers can be overcome, effectively enhancing the success of treatment programmes.

## Supporting information

**S1 Text. Tool: Interview guide addressed to parents/guardians of children 1–5 years old.**
(DOCX)

## Acknowledgments

We would like to thank all participants and the wider study team, including individuals who contributed to the overall study but not directly to this publication: Laura Roth, Dianne Verhoeven, Paskari Odoi, Joy Anyango, Joanita Bananuka, Zakia Karungi, Sarah Nankasa, Irene Kagoye, Peter Mutebe, Tabitha Naikesa, Tasibwire Brian Mugabi, John Bosco Asiimwe and the Pediatric Praziquantel Consortia.

## Author contributions

**Conceptualization:** Lisa Sophie Reigl, Mary Amuyunzu-Nyamongo, Andrea Buhl, Eveline Hürlimann, Suzanne Lavry, Janet Masaku, Nora Monnier, Peter Steinmann, Alain Toh, Isabelle L Lange.

**Data curation:** Lisa Sophie Reigl, Maxson Kenneth Anyolitho, Marie Frese, Isabelle L Lange.

**Formal analysis:** Lisa Sophie Reigl, Marie Frese, Isabelle L Lange.

**Funding acquisition:** Stella Neema, Mary Amuyunzu-Nyamongo, Andrea Buhl, Jennifer Burrill, Eveline Hürlimann, Nora Monnier, Alice Sereti Sinkeet, Peter Steinmann, Andrea S Winkler.

**Investigation:** Lisa Sophie Reigl, Maxson Kenneth Anyolitho, Stella Neema, Isabelle L Lange.

**Methodology:** Lisa Sophie Reigl, Stella Neema, Mary Amuyunzu-Nyamongo, Andrea Buhl, Djouquou Alexise Gnahore, Suzanne Lavry, Janet Masaku, Alice Sereti Sinkeet, Alain Toh, Andrea S Winkler, Isabelle L Lange.

**Project administration:** Isabelle L Lange.

**Supervision:** Stella Neema, Ashley Preston, Orkan Okan, Andrea S Winkler.

**Validation:** Stella Neema.

**Visualization:** Lisa Sophie Reigl.

**Writing – original draft:** Lisa Sophie Reigl.

**Writing – review & editing:** Lisa Sophie Reigl, Maxson Kenneth Anyolitho, Stella Neema, Mary Amuyunzu-Nyamongo, Andrea Buhl, Jennifer Burrill, Marie Frese, Djouquou Alexise Gnahore, Eveline Hürlimann, Suzanne Lavry, Janet Masaku, Nora Monnier, Ashley Preston, Alice Sereti Sinkeet, Peter Steinmann, Alain Toh, Orkan Okan, Andrea S Winkler, Isabelle L Lange.

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
