## [Decision Letter · Decision Letter 0]

30 Sep 2024

Dear Ms Reigl,

Thank you very much for submitting your manuscript "Socio-cultural and structural barriers influencing parents’ knowledge and access to information on schistosomiasis in children around Ugandan Lakes" for consideration at PLOS Neglected Tropical Diseases. As with all papers reviewed by the journal, your manuscript was reviewed by members of the editorial board and by several independent reviewers. The reviewers appreciated the attention to an important topic. Based on the reviews, we are likely to accept this manuscript for publication, providing that you modify the manuscript according to the review recommendations. 

The authors have contributed an important work towards the potential for schistosomiasis control in two endemic areas of Uganda, but additional clarity is needed in the description of the data collection, cleaning, and analytical methods, including justification for the approaches used and sample sizes for testing or exploring specific hypotheses or objectives. 

Sincerely,

Suzan CM Trienekens, PhD

Academic Editor

Amy Gilbert

Section Editor

Reviewer's Responses to Questions

**Key Review Criteria Required for Acceptance?**

**Methods**

-Are the objectives of the study clearly articulated with a clear testable hypothesis stated?

-Is the study design appropriate to address the stated objectives?

-Is the population clearly described and appropriate for the hypothesis being tested?

-Is the sample size sufficient to ensure adequate power to address the hypothesis being tested?

-Were correct statistical analysis used to support conclusions?

-Are there concerns about ethical or regulatory requirements being met?

Reviewer #1: The rationale for choosing villages within the two districts is not clear. How large are the districts of Bugiri and Hoima? How much of the district populations reside in five selected villages in the Bugiri and the ten villages in Hoima? How were these villages selected and what can be said about these villages in terms of the target population that the investigators have attempted to reach in this study? Statistical concerns do not apply to a qualitative study in the same way they would to a quantitative study, but the question of who is included/excluded in the study population has bearing on how the research question is answered and the conclusions that can drawn. The process of selecting the study population should be outlined in more detail, with justification/rationale provided for the choices made. 

How did the in-depth interviews, key informant interviews and focus group discussions differ in terms of the data collection instrument? Were different questions asked? Or the same questions asked differently? How did any differences get accounted for in the data analysis stage? Since there were different data collection modalities and different types of participants, some description of how those modalities differed in terms of both data collection and analysis is warranted.

Reviewer #3: • In the methods (2.5), you state that transcripts were ‘where necessary, edited’ – please include a brief sentence on what they were edited for/in what way they were edited. 

• What does s.b. stand for in Table 2? Can this be spelt out somewhere? 

• Table 3 captures the key themes really clearly and succinctly. A great addition to the paper. Could you expand ‘Correctness of information provided dependent on knowledge of the parents’ information providers’ a little bit?

**Results**

-Does the analysis presented match the analysis plan?

-Are the results clearly and completely presented?

-Are the figures (Tables, Images) of sufficient quality for clarity?

Reviewer #1: Please include a summary of the characteristics of the study villages (schistosomiasis burden; proximity to lake, school, market; community infrastructure observed) at the beginning of the results section. A high-level description of the data (e.g., how much audio was recorded and how many pages of transcripts did that produce in the end?) would also be helpful so readers can appreciate the amount of enormous amount of time and effort that goes into both collecting and analyzing qualitative data. 

In a study that is so focused on supporting the implementation of a treatment program in preschool-aged children, I was surprised that the authors buried their findings on parents knowledge of treatment so deep into their results section (it’s on the SIXTH page of the results!). The results seem to be structured in sort of a ‘kitchen sink’ way, where every topic covered in interviews and focus groups is addressed very briefly. This doesn’t allow the authors much room for elaboration or the inclusion of more than a quote or two per topic. It also makes it hard for the reader to extract a key message from the study’s findings. Instead of trying to fit so much into so little space, I would encourage the authors to lead with their top-line findings (e.g., what are the most important three points you would communicate to government partners?) and expand on those in the results and discussion section. The material covered in the results and discussion can, in turn, inform what information is essential to include in the introduction. 

How many parents had experience with MDA programs for their school-aged children? Did those experiences influence their attitudes towards treatment for PSAC?

Reviewer #3: This is overall a clearly written paper articulating a set of important findings, well situated within relevant and supportive literature. The descriptions and use of photographs in the methods give the reader a strong sense of what the communities are like, presented succinctly. There is a really nice summary of the challenges on lines 486-493: these show how geographical location, education levels, and technology intersect. This is very clear, and it is good to see this coupled with expansion of content in Table 3. The findings reinforce the findings of other studies, but do an excellent job of why attention to wider context is important. 

My main critique (if critique is even a useful word here) is about the way that local ways of knowing are presented in the paper and the ethical implications of this for producing meaningful change. The below (and I apologise that it is so long) is intended as an invitation for reflection on how the language we use can both help and hinder efforts to produce meaningful change. They are an invitation to build on the important strengths of this paper only. 

The first of these is your distinction between an ‘IDI’ (in-depth interview) and ‘KII’ (presumably key informant interview). In-depth is a description of a methodological approach – that the interview seeks to achieve depth – and is an approach that can be taken with any interviewee. KII suggests that some people are more ‘key’ than others – and thus that those others (here, presumably community members, who are the experts in their own lives) are less, although most of the findings focus on their interviews. Is this a distinction you want to reinforce? Interestingly, in your presentation of findings, you do not reinforce, but indeed blur, it. For example, ‘A 48 year old other from Bugiri…’ quoted with ‘KII, mother, Bugiri (501)’ as the descriptor after the quote, or ‘KII, mother, Hamia (209)’ earlier on. Given that in your analysis there does not seem to be a meaningful distinction between KII and parental interviews, why set up this distinction at all? Or, alternatively reflect on what it means that at least some of your ‘key’ informants answered as parents? What does this tell you? 

I further invite you to reflect on the implications of describing a group or individual’s way of understanding how transmission works as ‘a lack of understanding’ (352), a ‘misconception’ (379, 480), ‘belief’, ‘misbelief’ or ‘false assumptions’ (408). Think here about how you have characterised Parker et al (2011) in your discussion of the literature (151-154): you draw on this text to argue ‘local understandings of NTDs did not mirror biomedical knowledge’. Yet, in the cited paper, Parker et al do not argue that ‘local understandings of NTDs did not mirror biomedical knowledge’ but speak of ‘divergence between local and biomedical understandings’. This is a subtle but powerfully important difference – biomedical and local frames of understanding are different rather than one being false or lacking. When we as scholars and scientists speak to people to understand their world, and then articulate that to others what does it mean to describe those understandings as false, misconceptions, or lacking? I therefore advise you to look again at Parker et al 2011 and note the way they use language. You may also find Farmer’s 1990 ‘Sending Sickness’ useful – this is an excellent example of how multiple understandings of the cause of a sickness can circulate within a single community. If we want to make change through understanding, can we do that while calling our interlocutors’ understandings wrong? 

Related to this I notice is “bewitch” in inverted commas. Again, I invite you to reflect on the ethical implications of this punctuation choice: it implies that you do not take seriously bewitchment as a transmission mechanism. There is an argument in anthropology that reality is that which produces effects in the world, and bewitchment as a means of transmission (whether we or anyone else believes that bewitching is ‘real’ or not) produces effects in the world (again, Farmer’s ‘Sending Sickness’ and also AIDS and Accusation illustrate this beautifully). People seek out particular kinds of treatment (and perhaps not others) based on what they understand to be the cause. The preventative methods proposed for avoiding kidaada in the paper are entirely consistent with understandings of how kidaada is caused as presented in the paper. There are thus no ‘misconceptions’ (411) about prevention here – what there is, however, is a difference between how different groups (ie you and the community you seek to describe) understand schistosomiasis is transmitted, how kidaada is transmitted and the relationship between schistosomiasis and kidaada. What are the implications, then, of taking seriously the frames applied by the communities we wish to work with? Consequently, I was fascinated that the solution around lines 808-810 does not include engaging with and taking seriously witchcraft. If at least 1/3 people use it as a form of explaining causality and meaning-making, why not engage with it and take it seriously? How can you push your recommendations beyond simply ‘recognis[ing] the sensitivity of the subject’? Do we want to work with localised forms of sense-making or seek to replace them? What are the ethical implications of (either) answer to this question? 

For this, it may be worth thinking about how you communicate some of the complexity around trying to talk about a singular thing (‘schistosomiasis’, ‘bilharzia’) when ‘it’ clusters around different kinds of overlapping symptoms and causes. You do this beautifully in this paper. Your work here is important and makes visible the way in which what schistosomiasis is in these communities (and between these communities and research communities) is not simple or consistent. This is a real strength of the paper, as this complexity is hard to communicate and you have done so succinctly and effectively. I invite you to build on this strength in relation to the above. 

I was particularly struck by your presentation of participants who did not use PZQ. Resistance to PZQ in themselves seems important for understanding (barriers to uptake) in a paediatric formulation. You cite one participant who was clearly uncomfortable about telling you why they did not take it in the results and go on to say ‘Community members with different opinions may not always have opened up about them in interviews’ (474): did any community members tell you why they did not use PZQ? If not, what does reluctance to say why not tell you? Did people discuss making different choices about PZQ use for themselves or their children (ie refuse it themselves, but support their children taking it or vice versa)? I thus recommend that if there were any participants who did tell you why they did not take PZQ that you include an analysis of those people’s stated reasons; if they did not, and people who told you did not could or would not tell you why, keep this important point in, and reflect on it in the discussion. You can push your analysis beyond ‘shame’ (799) here. None of the material presented supports the idea that it is ‘shame’ that drove their non-disclosure, however what that quote does show is that when pressed some people who don’t take PZQ they don’t want to tell you why. Given the ways in which explanations for non-use are presented as something that others do in the rumours section, are there other ways of making sense of this than individualised feelings of shame? Great note of socially desirable responding though. Building on strengths in the paper, the paragraph on reasons why other people refuse PZQ/other vaccines blurs into the paragraph on impacts of new fishing regulation and enforcement. This very clearly shows how choices made about medication are not made in isolation from wider relations of power and larger political and economic forces. This is excellent. 

Related to this this, the discussion on rumours is very important. This really adds to the paper. However, in light of the invitation to explore more culturally relativist perspectives above, is it worth considering how what you have described as ‘rumours’ could also be understood as articulations of people’s fears? A ‘rumours’ framing has implicit within it that if only people were provided with the ‘facts’, different choices would be made. However, framing this through fears opens up different forms of engagement that start from and take seriously individuals’ and communities’ own understandings of the world. None of the above are recommendation but rather an invitation to pay attention to how the language we use can reinforce or challenge hierarchical knowledge-relations, and create space to situate our responses and solutions in ways that work with the worlds our research participants inhabit. 

Figure 5 (ln 773) captures most but not all of the material covered in the paper: there’s an important, fundamental mistrust of VHTs’ low level of training/community insider identity in the paper that is important, but not quite captured by ‘trust or mistrust in certain professions and political parties’. I also wonder in what ways ‘level of education’ is a barrier to ‘accessing and understanding accurate health information’ – is literacy/numeracy required for this? Is the burden on those seeking to sensitise perhaps to adapt their tools the community, rather than a limitation of the community? You have not evidenced how ‘customs and habits’ function as a barrier in the paper, but more importantly than that, I invite you to consider the ethical implications of situating responsibility for intervention failure on the ‘customs and habits’ of the target population. Pot’s 2019 paper ‘INGO Behavior Change Projects: Culturalism and Teenage Pregnancies in Malawi’ conveys Fassin’s concept of ‘culturalism’ really clearly and I advise you read it and consider the ethical implications of presenting the culture of the target population as a barrier. 

Smaller points: • In line 599-600 you state that ‘it became clear that misbeliefs by VHTs and other community-leading figures contribute to parents’ false assumptions’. Given you state above that with the exception of teachers (who demonstrated a high level of knowledge in line with MoH/biomedical understandings), the same range of understandings and assumptions is shared across parents, community leaders and VHTs, can you evidence from your material that it is specifically causal and not simply a reflection of the wide range of ideas about symptoms, causes etc present in the community? If not, soften your language. 

• Be careful of extrapolating out from a single participant to make claims about culture: Can you use one person referring to the lake as a ‘toilet of ours’ to make a claim that this is a ‘culturally accepted’ place to defecate? Are there latrines? Are they clean? Are they affordable? What did you find in your observations? Is it only ‘culture’ that drives people to defecate at the lake? Can you evidence this? If not, be very careful about making claims about ‘culture’ as a driver of practice (again, Pot 2019 will likely be useful here). 

• The discussion of lack of trust in VHTs is a really important aspect of the paper: there is real subtlety of analysis here. There is thus an opportunity on line 634 to build on this: low literacy rates, restricted access to info, and ‘relying on the scarce information from VHTs’ does not capture the lack of trust in VHTs articulated earlier. Just a few additional words here would make a huge difference. 

• Could you build on ‘Training for VHTs and community leaders are necessary to increase their knowledge, especially as they are important information providers for parents’ (819-820)? The paper really beautifully shows that local community perceptions about VHTs as no more educated than everyone else etc really impacts on the ways their messages are received. You’ve shown really nicely that community leaders have powerful influence, even if political figure do not. Therefore can you make more of this here? 

• Finally, could the fact that even with knowledge, practices cannot necessarily change due to the material conditions of people’s lives be moved up? This fact – and the reality of witchcraft as a meaningful force in the world for a significant proportion of your participants – might be important framing for subsequent recommendations about sensitisation efforts.

**Conclusions**

-Are the conclusions supported by the data presented?

-Are the limitations of analysis clearly described?

-Do the authors discuss how these data can be helpful to advance our understanding of the topic under study?

-Is public health relevance addressed?

Reviewer #1: (No Response)

Reviewer #2: (No Response)

Reviewer #3: (No Response)

**Editorial and Data Presentation Modifications?**

Reviewer #1: (No Response)

Reviewer #2: (No Response)

Reviewer #3: (No Response)

**Summary and General Comments**

Reviewer #1: In this study, the authors describe an impressive qualitative investigation involving parents, health workers and community leaders in two schistosomiasis-endemic districts in Uganda. Their study anticipates an important event in schistosomiasis control, as the Ugandan government prepares to pilot the distribution of a newly approved pediatric formulation of the anti-helminthic drug praziquantel. The study aims to capture knowledge of schistosomiasis, its treatment and sources of information in this population. Overall, this paper merits publication but there are several shortcomings in the current version that will require major revision before it is acceptable for publication. In addition to the methods and results comments above, I have outlined requests below for additional background and contextual information 

It is never mentioned which schistosome species affect the study area. Similarly, the authors do not describe the basic details of the transmission cycle, despite it being a topic discussed in interviews. 

What does a “positive opinion” from the European Medicine Agency mean? What have clinical trials of arpraziquantel shown? What is the Ugandan government’s timeline for piloting drug distribution in the pediatric population? Incorporating answers to these questions in the Introduction section is important contextual information for the current study.

How do MDA programs tend to work in Uganda? My general understanding is that these programs tend to be school-based to reach school-aged children. After noting in line 221 that there are few public primary schools in the study sites, though, I’m wondering if that assumption is incorrect.

Reviewer #2: Please find and consider suggested revisions in pdf attached.

Reviewer #3: This review is, I appreciate, a lot. And I hope you are able to take it in the spirit that it is intended: which is that this is an important paper, with moments of wonderful subtlety, complexity and compassion. It tells an important story and makes an important argument well. The above is thus offered in the spirit of taking something very good and making it excellent – or at the very least, inviting you to reflect on the value of starting from the worldviews of the communities with which we work, rather than seeking to supplant them (which I am not sure is your intention, just perhaps an accidental by-product of language choices). You do such a good job of capturing the ways in which seemingly unconnected political events (eg changes in fishing regulation or teacher recruitment practices) profoundly shape and inform understandings, I wish merely to invite you to build on these excellent strengths by pushing yourselves to reflect on language and framings a little more. Overall, I have found this paper insightful and subtle and hope to see it in print soon.

PLOS authors have the option to publish the peer review history of their article (what does this mean? ). If published, this will include your full peer review and any attached files.

**Do you want your identity to be public for this peer review?** For information about this choice, including consent withdrawal, please see our Privacy Policy .

Reviewer #1: No

Reviewer #2: Yes: Taissa Alice Soledade Calasans

Reviewer #3: Yes: Lucy Pickering

Figure Files:

Data Requirements:

Reproducibility:

References

---

## [Decision Letter · Decision Letter 1]

26 Feb 2025

PNTD-D-24-00908R1Socio-cultural and structural barriers influencing parents’ knowledge and access to information on schistosomiasis in children around Ugandan LakesPLOS Neglected Tropical Diseases Dear Dr. Reigl, Thank you for submitting your manuscript to PLOS Neglected Tropical Diseases. After careful consideration, we feel that it has merit but does not fully meet PLOS Neglected Tropical Diseases's publication criteria as it currently stands. Therefore, we invite you to submit a revised version of the manuscript that addresses the points raised during the review process. *Editor's note: two reviewers had several suggested edits that could improve the reporting and interpretation of the study. We are returning this to you as a minor revision to give you an opportunity to review these comments and edit the manuscript to incorporate the suggested changes that you think will improve the manuscript.  Please submit your revised manuscript within 30 days Mar 28 2025 11:59PM. If you will need more time than this to complete your revisions, please reply to this message or contact the journal office at plosntds@plos.org. Please include the following items when submitting your revised manuscript:* A rebuttal letter that responds to each point raised by the editor and reviewer(s). You should upload this letter as a separate file labeled 'Response to Reviewers '. This file does not need to include responses to any formatting updates and technical items listed in the 'Journal Requirements' section below.* A marked-up copy of your manuscript that highlights changes made to the original version. You should upload this as a separate file labeled 'Revised Manuscript with Track Changes '.* An unmarked version of your revised paper without tracked changes. You should upload this as a separate file labeled 'Manuscript '. If you would like to make changes to your financial disclosure, competing interests statement, or data availability statement, please make these updates within the submission form at the time of resubmission. Guidelines for resubmitting your figure files are available below the reviewer comments at the end of this letter. We look forward to receiving your revised manuscript. Kind regards, Elizabeth CarltonSection EditorPLOS Neglected Tropical Diseases

Shaden Kamhawi

co-Editor-in-Chief

Paul Brindley

co-Editor-in-Chief

 **Journal Requirements:**

Please ensure that the funders and grant numbers match between the Financial Disclosure field and the Funding Information tab in your submission form. Note that the funders must be provided in the same order in both places as well.

 **Reviewers' comments:** Reviewer's Responses to Questions

**Key Review Criteria Required for Acceptance?**

**Methods** :

-Are the objectives of the study clearly articulated with a clear testable hypothesis stated?

-Is the study design appropriate to address the stated objectives?

-Is the population clearly described and appropriate for the hypothesis being tested?

-Is the sample size sufficient to ensure adequate power to address the hypothesis being tested?

-Were correct statistical analysis used to support conclusions?

-Are there concerns about ethical or regulatory requirements being met?

Reviewer #1: (No Response)

Reviewer #2: (No Response)

Reviewer #3: -Are the objectives of the study clearly articulated with a clear testable hypothesis stated?

The study objectives are clearly stated

-Is the study design appropriate to address the stated objectives?

YES

-Is the population clearly described and appropriate for the hypothesis being tested?

YES

-Is the sample size sufficient to ensure adequate power to address the hypothesis being tested?

YES

-Were correct statistical analysis used to support conclusions?

NA

-Are there concerns about ethical or regulatory requirements being met?

There is a section on ethical approvals. I recommend that this section not be called ‘ethical considerations’ (given there is no discussion of the ethical considerations of the study) and be retitled ‘ethical approvals’.

**Results** :

-Does the analysis presented match the analysis plan?

-Are the results clearly and completely presented?

-Are the figures (Tables, Images) of sufficient quality for clarity?

Reviewer #1: (No Response)

Reviewer #2: (No Response)

Reviewer #3: -Does the analysis presented match the analysis plan?

YES

-Are the results clearly and completely presented?

YES

-Are the figures (Tables, Images) of sufficient quality for clarity?

YES

**Conclusions** :

-Are the conclusions supported by the data presented?

-Are the limitations of analysis clearly described?

-Do the authors discuss how these data can be helpful to advance our understanding of the topic under study?

-Is public health relevance addressed?

Reviewer #1: (No Response)

Reviewer #2: (No Response)

Reviewer #3: -Are the conclusions supported by the data presented?

YES

-Are the limitations of analysis clearly described?

YES

-Do the authors discuss how these data can be helpful to advance our understanding of the topic under study?

YES

-Is public health relevance addressed?

YES

**Editorial and Data Presentation Modifications?**

Reviewer #1: (No Response)

Reviewer #2: (No Response)

Reviewer #3: 1. as noted above, given the section on ethics does not actually address ethical dimensions of the study, this should be retitled ‘ethical approvals’.

2. The quote on lines 474-482 do not actually evidence that not taking medications is taboo, although it does evidence that some people do not feel comfortable about it. There is a nice discussion of this in the discussion section – could the reference to taboo be removed here?

3. Would the discussion of language at the start of the findings be better moved to section 3.6.1?

4. On lines 689-690 you state that teachers who don’t speak the local languages have communication difficulties ‘with mostly illiterate or low-educated people’. The fact that incomers do not speak locally spoken languages has nothing to do with literacy or with educational attainment of the community they have arrived into. It may be that teaching takes place in English and so people with more schooling would have a shared language with incomers but please find a way to rephrase this so it does not accidentally read as blaming outsiders’ lack of local language on the educational attainment of the communities under study.

**Summary and General Comments** :

Reviewer #1: I am satisfied with the changes made to the manuscript

Reviewer #2: (No Response)

Reviewer #3: (No Response)

PLOS authors have the option to publish the peer review history of their article (what does this mean? ). If published, this will include your full peer review and any attached files.

**Do you want your identity to be public for this peer review?** For information about this choice, including consent withdrawal, please see our Privacy Policy .

Reviewer #1: No

Reviewer #2: No

Reviewer #3: **Yes: ** Lucy Rose Pickering

---

## [Editor Report · Decision Letter 2]

11 Apr 2025

Dear Ms Reigl,

We are pleased to inform you that your manuscript 'Socio-cultural and structural barriers influencing parents’ knowledge and access to information on schistosomiasis in children around Ugandan Lakes' has been provisionally accepted for publication in PLOS Neglected Tropical Diseases.

Best regards,

Elizabeth J. Carlton

Section Editor

Shaden Kamhawi

co-Editor-in-Chief

Paul Brindley

co-Editor-in-Chief

---

## [Editor Report · Acceptance letter]

Dear Ms Reigl,

We are delighted to inform you that your manuscript, "Socio-cultural and structural barriers influencing parents’ knowledge and access to information on schistosomiasis in children around Ugandan Lakes," has been formally accepted for publication in PLOS Neglected Tropical Diseases.

Best regards,

Shaden Kamhawi

co-Editor-in-Chief

Paul Brindley

co-Editor-in-Chief
